# UDP-Glycosyltransferases from the UGT344 Family Are Involved in Sulfoxaflor Resistance in *Aphis gossypii* Glover

**DOI:** 10.3390/insects12040356

**Published:** 2021-04-16

**Authors:** Kangsheng Ma, Qiuling Tang, Pingzhuo Liang, Jianhong Li, Xiwu Gao

**Affiliations:** 1Department of Entomology, China Agricultural University, Beijing 100193, China; txma1986@163.com (K.M.); tangqiuling1145@163.com (Q.T.); liangpingzhu@126.com (P.L.); 2Hubei Insect Resources Utilization and Sustainable Pest Management Key Laboratory, College of Plant Science and Technology, Huazhong Agricultural University, Wuhan 430070, China; jianhl@mail.hzau.edu.cn

**Keywords:** UDP-glycosyltransferase, insecticide resistance, sulfoxaflor, *Aphis gossypii*, RNAi

## Abstract

**Simple Summary:**

The cotton aphid, *Aphis gossypii* Glover, is a notorious pest in cotton and cucurbit fields. The control of *A. gossypii* has typically relied on the application of chemical insecticides. Sulfoxaflor is the first commercially available sulfoximine insecticide, which exhibits great efficacy against sap-feeding insect pests and has been applied as an alternative insecticide for controlling of *A. gossypii* in China. Consequently, *A. gossypii* quickly developed resistance to this insecticide. Hence, in this study, to clarify the potential detoxifying roles of UGTs (one of the phase II detoxification enzymes) in resistance of *A. gossypii* against sulfoxaflor, the synergistic effects of two synergists (sulfinpyrazone and 5-nitrouracil) against sulfoxaflor were investigated using the susceptible and laboratory-established sulfoxaflor resistant strain (SulR), and the expression levels of 15 UGT genes were determined by qRT-PCR. Furthermore, the involvement of highly upregulated UGTs in sulfoxaflor-resistant strain was functionally tested by RNA interference (RNAi). Our results suggest that overexpression of UGTs contributes to sulfoxaflor resistance in *A. gossypii*, which should be useful for understanding sulfoxaflor resistance mechanisms.

**Abstract:**

UDP-glycosyltransferases (UGTs) are major phase II detoxification enzymes that catalyze the transfer of glycosyl residues from activated nucleotide sugars to acceptor hydrophobic molecules and play very important roles in the biotransformation of various endogenous and exogenous compounds. Our previous studies demonstrated that UGTs participated in the detoxification of insecticides in *Aphis gossypii*. However, the potential roles of UGTs in *A. gossypii* resistance to sulfoxaflor are still unclear. In this study, two inhibitors of UGT enzymes, sulfinpyrazone and 5-nitrouracil, significantly increased the toxicity of sulfoxaflor to a resistant strain of *A. gossypii*, whereas there were no synergistic effects in the susceptible strain. Based on the transcriptome sequencing results, the expression levels of 15 UGTs were analyzed by quantitative real-time PCR, and we found that seven UGT genes were highly over-expressed in a sulfoxaflor-resistant strain compared to the susceptible strain, including *UGT344B4*, *UGT344C5*, *UGT344A11*, *UGT344A14*, and *UGT344L2*. Further suppressing the expression of *UGT344B4*, *UGT344C5*, and *UGT344A11* by RNA interference significantly increased the sensitivity of resistant aphids to sulfoxaflor, indicating that the overexpression of UGT genes is potentially associated with sulfoxaflor resistance. These results could provide valuable information for further understanding the mechanisms of insecticide resistance.

## 1. Introduction

The cotton aphid, *Aphis gossypii* Glover (Hemiptera: Aphididae), is a notorious pest in cotton and cucurbit fields throughout the world. It causes great economic losses both through directly feeding and indirectly by virus transmission and contamination of honeydew [1]. The control of *A. gossypii* has typically relied on the application of chemical insecticides [2,3,4]. Consequently, cotton aphids have evolved very high levels of resistance to a range of chemical insecticides, such as organophosphates, carbamates, pyrethroids, and neonicotinoids [5,6,7,8,9].

In the last few decades, synthetic organic insecticides have been widely used to control insect pests in agriculture and horticulture worldwide [10,11,12,13,14,15], and many insect pests have developed resistance to several types of chemical insecticides [16,17,18,19,20,21], which is a great challenge for controlling of these pests effectively. It is well known that insects have developed four types of resistance mechanisms to chemical insecticides: metabolic resistance, target-site resistance, penetration resistance, and behavioral resistance [22]. Among these mechanisms, the enhanced detoxification mediated by the overproduction of detoxifying enzymes, including cytochrome P450 monooxygenases (P450s), esterases (ESTs), and glutathione *S*-transferases (GSTs), could result in pest resistance to multiple insecticide classes [4,23,24,25,26,27]. For example, the overexpression of a carboxylesterase gene contributes to omethoate resistance in *A. gossypii* [23]. The overexpression of multiple P450 genes, especially *CYP6ER1*, has been found to be involved in the resistance of *Nilaparvata lugens* to imidacloprid, thiamethoxam, sulfoxaflor, and clothianidin [28,29,30,31,32,33]. The elevated activities of carboxylesterases confer resistance to organophosphates, pyrethroids, and carbamates in *Myzus persicae* [24,34,35,36]. Recently, the involvement of UDP-glycosyltransferases (UGTs) in secondary metabolism was also reported in several insect pests [37,38,39,40,41].

UGTs are a superfamily of enzymes found in animals, plants, fungi, and bacteria that catalyze the conjugation of a range of diverse small lipophilic compounds with sugars to produce glycosides, generating water-soluble products and resulting in the detoxification and elimination of their substrate [42,43,44,45,46]. At first, UGTs were found to play very important roles in the detoxification of plant secondary metabolites in insects [47,48,49,50]. Recently, several studies have demonstrated that UGTs might be associated with insecticide resistance [41,51,52,53,54,55]. For instance, the overexpression of *UGT2B17* (renamed *UGT33AA4*) is involved in chlorantraniliprole resistance in *Plutella xylostella* [40,41], and the upregulation of UGT genes was proved to be associated with imidacloprid resistance in *Diaphorina citri* [54]. Our previous studies also found that UGTs potentially contributed to imidacloprid resistance in *A. gossypii* [10,38]. Sulfoxaflor is a sulfoximine insecticide, which is widely used in the control of *A. gossypii* in China [56,57]. Our long-term resistance monitoring results show that field populations of *A. gossypii* collected from cotton fields in China had evolved low to moderate levels of resistance to sulfoxaflor. The further resistance mechanism study indicated that overexpression of multiple P450 genes contribute to sulfoxaflor resistance [4]. However, whether UGTs are involved in sulfoxaflor resistance in *A. gossypii* has not been determined.

In the present study, to clarify the potential roles of UGTs in resistance of *A. gossypii* to sulfoxaflor, the synergistic effects of two synergists (sulfinpyrazone and 5-nitrouracil) against sulfoxaflor were investigated by using the susceptible and laboratory established sulfoxaflor-resistant strain (SulR), and the transcriptional levels of 15 UGT genes were determined by qRT-PCR. Furthermore, the involvement of highly overexpressed UGTs in the sulfoxaflor-resistant strain was functionally tested by RNA interference (RNAi). Our results suggest that overexpression of UGTs can also contribute to the sulfoxaflor resistance in *A. gossypii*, which should be useful for understanding sulfoxaflor resistance mechanisms. In addition, these results might facilitate further study of the functions of UGTs in insecticide resistance.

## 2. Materials and Methods

### 2.1. Insects

Two strains of *A. gossypii* used in this study were established in the laboratory using the same original field population collected in 2016 from cotton fields in Xinjiang Uygur Autonomous Region, China. One strain was susceptible to sulfoxaflor (SS) and the other strain was resistant to sulfoxaflor (SulR), which was established from the Shawan population by continual selection with increased concentration of sulfoxaflor based on the LC_50_ values [4]. All aphids were reared on cotton seedlings, *Gossypium hirsutum* (L.), in the laboratory at 22 ± 1 °C, with 60% relative humidity and a photoperiod of 16:8 h (light:dark).

### 2.2. Chemicals

Sulfoxaflor (97.9%) was obtained from Dow AgroSciences (Indianapolis, Indiana, USA); two inhibitors of UGT enzymes, sulfinpyrazone (SUL) and 5-nitrouracil (5-NU), and Triton X-100 were purchased from Sigma-Aldrich (Saint Louis, MO, USA). Other chemicals and reagents used in this study were analytical grade reagents.

### 2.3. Toxicity Bioassays

The toxicities of the sulfoxaflor to SS and SulR populations were evaluated by using the leaf-dipping method [58] with slight modifications [4]. The stock solution of sulfoxaflor was prepared in acetone and adjusted to the desired concentrations by serial dilution with distilled water containing 0.05% (*v*/*v*) Triton X-100 for the bioassays. Fresh cotton leaves were cut into 20 mm diameter leaf discs using a sharpened steel punch. The leaf discs were then dipped in the desired concentration of sulfoxaflor or in 0.05% (*v*/*v*) Triton X-100 water for 15 s as a control. The treated leaf discs were placed in the shade and allowed to air dry, and then placed upside down onto the agar beds (1.5 mL of 2% agar) in the 12-well cell culture plate. Finally, healthy apterous adult aphids were carefully transferred onto the leaf discs and covered with Chinese art paper to prevent escape. All bioassays were carried out in the laboratory at 22 ± 1 °C with a photoperiod of 16:8 h (light:dark), the same as the insect rearing. The treatment for each concentration was performed with three replicates, and at least 30 aphids were used for each replicate. The mortality was recorded at 48 h after treatment. The LC_50_ values were calculated by probit analysis using POLO Plus 2.0 statistical software (LeOra Software Inc., Berkeley, CA, USA).

### 2.4. Synergism Bioassays

To reveal the potential role of UGTs in metabolic resistance against sulfoxaflor, the synergistic effects of sulfinpyrazone (SUL) and 5-nitrouracil (5-NU) were investigated using the bioassay method described above. The maximum sublethal doses of SUL and 5-NU for the susceptible strain were determined using the above-mentioned bioassay method, and the maximum doses that led to zero mortality in the susceptible strain were adopted as the maximum sublethal concentrations in the present study. Healthy apterous adult aphids were exposed to cotton leaf discs that were treated with SUL or 5-NU (at the maximum sublethal concentration) and the sulfoxaflor mixtures. The final concentration of SUL and 5-NU were 80 μg mL^−1^. The mortality was recorded at 48 h after treatment. The probit analysis was conducted using POLO Plus 2.0 statistical software and the synergistic ratio (SR) was calculated by dividing the LC_50_ values without the synergist by the LC_50_ values with synergist.

### 2.5. Quantitative Real-Time PCR and Data Analysis

The relative expression levels of UGT genes in the SS and SulR strains were determined using quantitative real-time PCR (qRT-PCR). Total RNA of cotton aphids was isolated using TRIzol^®^ reagent (Invitrogen, Carlsbad, CA, USA) following the manufacturer’s instructions, and first-strand cDNA was synthesized from 1.0 ug total RNA using the PrimeScript RT reagent kit with gDNA Eraser (Takara Biotechnology, Dalian, China), and stored at −20 °C. qRT-PCR was performed on an Applied Biosystems 7500 Real-Time PCR system (Applied Biosystems, Foster City, CA, USA) using SYBR^®^ Premix Ex Taq™ (Tli RNaseH Plus) (Takara Biotechnology, Dalian, China). The qRT-PCR reactions and the thermocycling program were described in our previous publications [4,59]. The experiment was conducted with three technical replications and three independent biological replicates. The housekeeping genes elongation factor 1 alpha (*EF1α*) and beta actin (*β-ACT*) were used as internal reference genes in this study [60]. The relative gene expression was calculated using the 2^-∆∆*C*t^ method [61].

### 2.6. UGT Genes Silencing and Bioassays

Primer design and synthesis methods for dsRNA production were described in our previous publication [4]. The gene fragments of four highly overexpressed UGT genes (*UGT344B4*, *UGT344C5*, *UGT344A11*, and *UGT344L2*) were amplified using specific primers conjugated with the T7 RNA polymerase promoter (Table 1). The products of RT-PCR were used as templates for dsRNA synthesis using the MEGAscriptRNAi kit (Ambion, Austin, USA). The dsRNA of *GFP* was synthesized under the same conditions as the UGT genes. For the dsRNA feeding experiments, dsRNA was added to an artificial diet (0.5 mol L^−1^ sterile sucrose solution) at a final concentration of 100 ng μL^−1^. The artificial diet containing DEPC water and 100 ng μL^−1^
*dsGFP* were used as the control. At least fifty healthy apterous adults of SulR strain were used for each dsRNA feeding experiments. The experiments were performed in triplicate. To determine the efficiency of dsRNA knockdown of target UGT genes, the living aphids were collected after feeding for 48 h, and then the samples were used for qRT-PCR analysis. To assess the sensitivity of *A. gossypii* to sulfoxaflor after RNAi of UGT genes, fifty resistant adult aphids that fed on dsRNA incorporated artificial diets for 48 h were transferred onto cotton leaf discs that had been treated with a sulfoxaflor solution of 130 mg L^−1^ (approximate concentration of LC_50_ value of sulfoxaflor to SulR). The mortality was recorded after exposure to sulfoxaflor for 48 h. Each treatment was replicated in triplicate.

### 2.7. Data Analysis

The gene expression data were analyzed using Student’s *t*-test for significant differences between the sulfoxaflor-susceptible and -resistant strains, and *p* < 0.05 was considered to be statistically significant. One-way analysis of variance (ANOVA), with Tukey’s multiple comparisons, was used to compare the RNAi silencing efficiency and the effect of the dsRNA treatments on insecticide susceptibility. All statistical analysis was completed using the GraphPad InStat 3.0 software (GraphPad Software, San Diego, CA, USA).

## 3. Results

### 3.1. Synergism of Sulfinpyrazone and 5-Nitrouracil

The synergisms of SUL and 5-NU to sulfoxaflor were evaluated by the leaf-dipping method, and the results are summarized in Table 2. Both SUL and 5-NU exhibited no significant synergism against sulfoxaflor in the susceptible strain (SS), while these two synergists significantly increased the toxicity of sulfoxaflor in the SulR strain by 2.01- and 3.78-fold, respectively (Table 2) (*p* < 0.05). It suggests that UDP-glycosyltransferases may be associated with the resistance of *A. gossypii* against sulfoxaflor.

### 3.2. Expression Profiles of Aphis Gossypii UGT Genes in the Susceptible and Resistant Strains

Thirty-one UGT genes from the SS and SulR strains of *A. gossypii* were identified based on our previous transcriptomic sequencing results, and six of them were overexpressed in SulR strain (Appendix A). To investigate which UGT genes may be involved in sulfoxaflor detoxification, the mRNA levels of 15 UGT genes in the susceptible and resistant strains were analyzed by quantitative real-time PCR. The results show that the transcripts of seven UGT genes were significantly upregulated in the SulR strain compared with the SS strain (*p* < 0.05) (Figure 1), among which four UGT344 family genes were increased more than two-fold. Specifically, the levels of *UGT344B4*, *UGT344C5*, and *UGT344A11* were increased to 4.40-, 4.09-, and 3.02-fold, respectively (Figure 1). In contrast, the transcripts levels of *UGT342A2* and *UGT350C2* were downregulated in the SulR strain compared with the SS strain (Figure 1).

### 3.3. Effects of UGT Gene Suppression on the Sensitivity of Aphis Gossypii to Sulfoxaflor

To evaluate the functional roles of overexpressed UGT genes in *A. gossypii* resistance against sulfoxaflor, the expression levels of four UGT344 family genes were depressed by RNAi in the sulfoxaflor resistant strain, and the susceptibility to sulfoxaflor was determined after dsRNA feeding treatments. After resistant cotton aphids were fed on dsRNA incorporated diet for 48 h, the expression levels of *UGT344B4*, *UGT344C5*, *UGT344A11*, and *UGT344L2* were reduced to 0.68-, 0.53-, 0.56-, and 0.64-fold compared to that of the control, which only contained *dsGFP* (Figure 2). The toxicity of sulfoxaflor to aphids significantly increased after the treatment by feeding the *dsUGT344B4*, *dsUGT344C5*, and *dsUGT344A11* and the mortalities were 85.9%, 84.9%, and 87.8%, respectively, which were significantly higher than that of the control (61.0% for DEPC water treatment and 56.3% for *dsGFP* treatment) at the diagnostic dose of sulfoxaflor (Figure 3). While feeding on *dsUGT344L2* for 48 h did not significantly increase the susceptibility of resistant adult aphids to sulfoxaflor, the mortality was 67.8% (Figure 3).

## 4. Discussion

Cotton aphids have developed high levels of resistance to a wide range of chemical insecticides in China due to the long-term and extensive application of traditional insecticides in the fields [8,10]. Sulfoxaflor has been used as an alternative insecticide for controlling of *A. gossypii* in China because of its high efficacy against sucking insect pests [62,63]. However, our previous study demonstrated that *A. gossypii* possessed high risk for developing resistance to sulfoxaflor, and further study of resistance mechanisms using a laboratory established resistant strain indicated that up-regulation of multiple P450 genes contributed to sulfoxaflor resistance [4]. In this study, to clarify the roles of phase II enzymes in the UGT family in sulfoxaflor resistance, both UGT inhibitors 5-nitrouracil and sulfinpyrazone were used in a synergism assay. The results show that two synergists significantly increased the toxicity of sulfoxaflor against resistant aphids, but showed no synergism effects against the susceptible strain (Table 2). This is in accordance with our previous study showing that 5-nitrouracil can increase the toxicity of sulfoxaflor in laboratory and two field populations [10], suggesting that enhanced UGT enzyme activities may contribute to the sulfoxaflor resistance in *A. gossypii*.

Although UGTs might be associated with sulfoxaflor resistance in *A. gossypii*, the specific genes that could play very important roles in this process are still unknown. Therefore, to determine the potential involvement of UGT genes in *A. gossypii* resistance to sulfoxaflor, the expression levels of 15 UGT genes were determined by using qRT-PCR, and the results show that seven of them were significantly overexpressed in the SulR strain compared with the susceptible strain. This is similar to some other studies in insects [41,54]. For example, Tian et al. found that 14 UGT genes were significantly upregulated in an imidacloprid-resistant strain of *D. citri* [54]. Similarly, in *P. xylostella*, Li et al. found that 10 UGTs were overexpressed in a multi-resistant field population (BL) compared to the SS population [40]. Interestingly, we observed that five overexpressed UGT genes belong to the UGT344 family, three of which increased expression more than 3.0-fold. Similar overexpression levels of these genes were observed in an imidacloprid-resistant strain of *A. gossypii* [38]. This suggests that more than one UGT gene in a family might participate in the detoxification of sulfoxaflor in *A. gossypii*.

To further confirm the function of overexpressed UGT genes in *A. gossypii* resistance to sulfoxaflor, the effects of the depressed expression of four highly upregulated UGT genes on the susceptibility to sulfoxaflor were evaluated by RNAi. Finally, we found that suppression of the expression levels of *UGT344B4*, *UGT344C5*, and *UGT344A11* could significantly increase the sensitivity of sulfoxaflor resistant cotton aphids to sulfoxaflor, suggesting that these three UGT genes might participate in the detoxification process of sulfoxaflor in *A. gossypii*, and contribute to the development of sulfoxaflor resistance. Similar results were observed in *Tetranychus cinnabarinus* that silencing of *UGT201D3* resulted in significantly increased susceptibility to abamectin in a resistant strain [39]. In the present study, we found that *UGT344B4* plays a very important role in detoxifying sulfoxaflor, which is also confirmed in imidacloprid and thiamethoxam-resistant strains where suppression of *UGT344B4* resulted in significantly increased susceptibility of *A. gossypii* to imidacloprid, bifenthrin [10,38], and thiamethoxam [55]. Both these results suggest that *UGT344B4* might be involved in the detoxification of neonicotinoid, sulfoximine, and pyrethroid insecticides in *A. gossypii*. However, our results indicate that overexpression of *UGT344C5* and *UGT344A11* contributes to sulfoxaflor resistance in *A. gossypii*, which is different from the results obtained in imidacloprid and thiamethoxam-resistant strains [38,55]. This difference may be due to the differential detoxification mechanisms in *A. gossypii* against various types of insecticides. It is worth pointing out that our present results fail to provide direct evidence of the metabolism of sulfoxaflor by UGTs. Therefore, whether overexpressed UGTs can metabolize sulfoxaflor needs further functional research.

## 5. Conclusions

Although our previous study demonstrated that elevated detoxification by P450 enzymes caused by overexpression of multiple P450 genes could be the main metabolic resistance mechanism of sulfoxaflor in *A. gossypii*, UGTs, as one type of important phase II metabolic enzymes, were also associated with the sulfoxaflor resistance in *A. gossypii*. These results should be useful for helping gain a better understanding of the sulfoxaflor resistance mechanisms in *A. gossypii* and provide valuable information for further understanding the function of UGTs in insects against insecticide stress.

## Figures and Tables

**Figure 1 insects-12-00356-f001:**
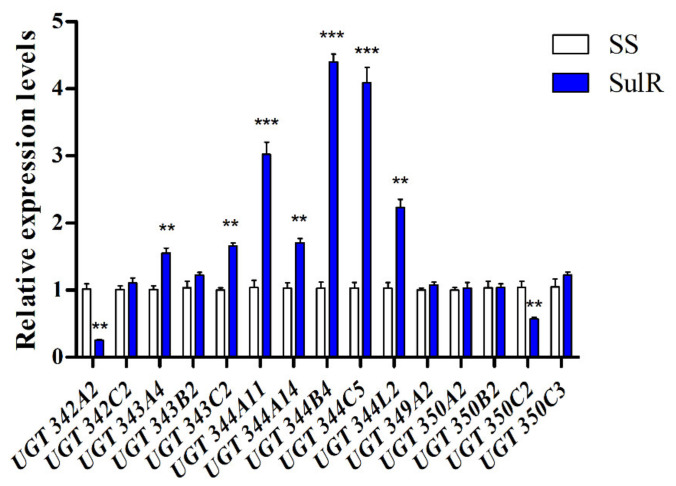
Relative expression levels of UGT genes in adult aphids of the susceptible and sulfoxaflor resistant strains. Data are presented as the mean ± SD for at least three independent replicates. ** indicates significant differences as analyzed by Student’s *t*-test (*p* < 0.05); *** indicates significant differences as analyzed by Student’s *t*-test (*p* < 0.01).

**Figure 2 insects-12-00356-f002:**
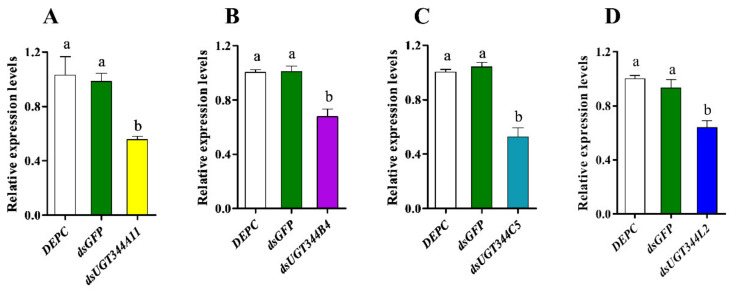
The transcription levels of UGT genes in *Aphis gossypii* fed with corresponding dsRNA (100 ng μL^−1^). The transcription levels of *UGT344A11* (**A**), *UGT344B4* (**B**), *UGT344C5* (**C**), and *UGT344L2* (**D**) in *A. gossypii*. The DEPC water and *dsGFP* (100 ng μL^−1^) treatments were used as control. Different lowercase letters (a, b) with the bars indicated that the means are significantly different according to one-way ANOVA, followed by Tukey’s multiple comparison test (*p* < 0.05).

**Figure 3 insects-12-00356-f003:**
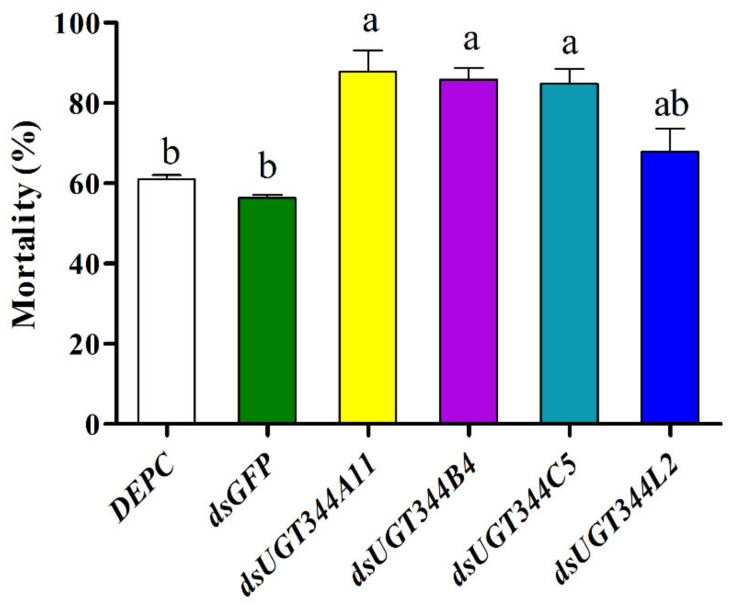
Effects of knockdown of UGT genes on the mortality of resistant aphids after treatment with the LC_50_ of sulfoxaflor for 48 h. Different lowercase letters (a, b) with the bars indicated that the means are significantly different according to one-way ANOVA, followed by Tukey’s multiple comparison test (*p* < 0.05).

**Table 1 insects-12-00356-t001:** Primers used for qRT-PCR analysis and dsRNA synthesis.

Primer Name	Forward Primer (5′–3′)	Reverse Primer (5′–3′)
*EF1α*	GAAGCCTGGTATGGTTGTCGT	GGGTGGGTTGTTCTTTGTG
*β-Actin*	GGGAGTCATGGTTGGTATGG	TCCATATCGTCCCAGTTGGT
*UGT344C5*	GCCGAATCCAGCAACAGTAT	TTCATGAACACCAACGACGG
*UGT344B4*	GGTTCGTGGGTCACTACTCC	TTGCCCATCTAGTATCTTCTCA
*UGT344D6*	GTCAGCCCATCTATTATCTTCC	GGCGGGTTTCAGGTGTAT
*UGT344L2*	TCCGCCGTTCCCAAGAC	CCACCGACACTAACAACATTCG
*UGT344A11*	GCCAAGCACGGAAGTCA	ACGCACTCGGACACCAG
*UGT344A14*	GGGACTTGAAGGTTAGGG	ATCGGTGACGGAATGAC
*UGT343A4*	TCATAACTCACGGAGGATTG	GCACTTCTTTGACGGCATT
*UGT343B2*	CCGTCAATGGTCTGGGTC	TGAGCGTTCATCAGCGTTA
*UGT343C2*	ATCCGTCCACTTTACCA	TGAATCCCACTTCCACA
*UGT342A2*	CAAAGCCACTGTTGCCTAAT	AATACGCTGGTGCTGTTTC
*UGT342C2*	AAACGACGCTCAACTAACCA	GGAGCCGAGCAATTCTGT
*UGT349A2*	CGGTGGACTGTTAGGGGTA	CGCATTTATAGCGTAACTGTCA
*UGT350A2*	CACAGTGTTGAAGAGGCAGT	AGCAGCTCCTCTAGATTCCA
*UGT350B2*	CATCTATTCCAAATGCTGGTG	TGACGGTCGTGTCTCCC
*UGT350C2*	AAAATGCCCAAGGAAACAG	GGGAACTCCGTGATAGACG
*UGT350C3*	GTGTCGCAGTGGCAAGG	CGTTCTGGAGCATCGTCT
344C5-RNAi	taatacgactcactatagggAGCACAAGTACCTCAGAGAGT	taatacgactcactatagggACAACTGATTCTGCTGGTGAC
344B4-RNAi	taatacgactcactatagggACGATGAGTAGAATGCTGTGC	taatacgactcactatagggGACTTGCCGGTTCGATTGTA
344L2-RNAi	taatacgactcactatagggTATGAGTGCTGTGCTTCGAG	taatacgactcactatagggATTGTTGACACCGTTGCTGG
344A11-RNAi	taatacgactcactatagggTGGACATGAACGGATGGTGA	taatacgactcactatagggCGTGCCGATTCAGTGATGAA

**Table 2 insects-12-00356-t002:** Synergistic effects of 5-NU and SUL on the toxicity of sulfoxaflor in the susceptible and sulfoxaflor resistant strains of *Aphis gossypii*.

Strains	Insecticide + Synergist	Slope ± SE ^a^	LC_50_ (95% CL) (mg L^−1^) ^b^	*χ^2^*	*df*	SR ^c^
SS	Sulfoxaflor	1.18 ± 0.15	0.40 (0.24–0.57)	9.37	16	
	Sulfoxaflor +5-NU	0.93 ± 0.14	0.32 (0.14–0.51)	4.84	16	1.28
	Sulfoxaflor + SUL	0.96 ± 0.14	0.35 (0.17–0.58)	18.36	19	1.14
SulR	Sulfoxaflor	1.05 ± 0.16	131.30 (90.59–209.20)	11.38	16	
	Sulfoxaflor +5-NU	1.09 ± 0.14	65.40 (47.34–90.29)	13.63	16	2.01
	Sulfoxaflor + SUL	1.75 ± 0.33	34.74 (13.58–52.39)	23.69	16	3.78

^a^ SE: standard error; ^b^ CL: confidence limits; ^c^ SR: synergism ratio = LC_50_ of sulfoxaflor/LC_50_ of sulfoxaflor with synergist.

## Data Availability

Not applicable.

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
