# Peer review of "UDP-Glycosyltransferases from the UGT344 Family Are Involved in Sulfoxaflor Resistance in Aphis gossypii Glover"

_insects, 2021, doi:10.3390/insects12040356_

Round 1

Reviewer 1 Report

While the methods and results of this paper are scientifically sound, it needs to undergo extensive english editing before it can be recommended for publication. Some other major/minor changes that authors need to make to improve the quality and readability of this paper are outlined below.

Major and minor comments:

  1. Why is the scientific name Aphis gossypii not italicized throughout the paper. Is it the journal policy not italicize scientific names?
  2. Lines 45-52: Instead of providing resistance mechanism examples for other insect pests, the authors should specifically discuss A. gossypii resistance mechanisms reported in previously published papers.
  3. Lines 69-70: Authors do not have to state that the role of P450s in sulfoxaflor resistance has not been previously determined. This is because this paper is specifically related to UGT's and authors have not investigated P450 expression in the resistant strain.
  4. Lines 83-86: For clarity, it is important that authors describe how the resistant strain was established in 1-2 or more sentences. Was is selected with sulfoxaflor in the laboratory? It is not sufficient to merely cite the reference.
  5. Section 2.5: How did the authors identify UGT genes used in this study. Did they rely on a previously sequenced transcriptome or genome? This information needs to be included in this section along with appropriate reference(s).
  6. Section 2.7: Why did the authors use student's t test for gene expression analysis and ANOVA + Tukey's test for RNAi experiments. The same ANOVA followed by Tukey's approach would have been a robust one for gene expression/ qRT-PCR data.
  7. Section 3.2: 3-4 fold is a modest increase in expression of UGT genes. Authors need to discuss the relevance of this modest increase in the discussion section. Specifically they need to discuss how much increase in UGT gene expression was reported in other studies/ other insect species.
  8. Lines 244-246: What is the significance of CYP6AE gene cluster with the role of UGT's in resistance? This needs to be explained in detail in the discussion section.

Author Response

While the methods and results of this paper are scientifically sound, it needs to undergo extensive English editing before it can be recommended for publication. Some other major/minor changes that authors need to make to improve the quality and readability of this paper are outlined below.

Major and minor comments:

1. Why is the scientific name Aphis gossypii not italicized throughout the paper. Is it the journal policy not italicize scientific names?

Response: Thanks for your comments. In our original submission the scientific names of Aphis gossypii and other insect species were in italics, while the editorial office have made some formatting changes to our original submission that led scientific names were not italicized. In the current version, all scientific names of insect species were written in italics throughout the manuscript.

2. Lines 45-52: Instead of providing resistance mechanism examples for other insect pests, the authors should specifically discuss A. gossypii resistance mechanisms reported in previously published papers.

Response: Thanks for your comments. In this part, two examples about A. gossypii resistance mechanisms have already cited in the resistance mechanism examples provided here (references 37 and 38 are two examples that indicated the involvement of UGT in A. gossypii resistance to insecticides). According to your suggestion the first example has been replaced by a A. gossypii resistance related example (Line 46-47).

3. Lines 69-70: Authors do not have to state that the role of P450s in sulfoxaflor resistance has not been previously determined. This is because this paper is specifically related to UGT's and authors have not investigated P450 expression in the resistant strain.

Response: Thanks for your comments. This sentence was revised to “However, whether UGTs are involved in sulfoxaflor resistance in A. gossypii has not been determined”, and the redundant description about P450 was deleted in this revision (Line 70-71).

4. Lines 83-86: For clarity, it is important that authors describe how the resistant strain was established in 1-2 or more sentences. Was is selected with sulfoxaflor in the laboratory? It is not sufficient to merely cite the reference.

Response: Thanks for your comments. The description of aphid strains used in this study was revised and more information was added in the section of materials and methods (Line 87-90).

5. Section 2.5: How did the authors identify UGT genes used in this study. Did they rely on a previously sequenced transcriptome or genome? This information needs to be included in this section along with appropriate reference(s).

Response: Thanks for your comments. The UGT genes were identified from our previously sequenced transcriptome, which was indicated in the results (section 3.2) with a sentence “thirty-one UGT genes from the SS and SulR strains of A. gossypii were identified based on our previous transcriptomic sequencing results” (Line 183-184), in the section 2.5 we do not think it is necessary to include this information since this section mainly described the methods about the qRT-PCR analysis.

6. Section 2.7: Why did the authors use student's t test for gene expression analysis and ANOVA + Tukey's test for RNAi experiments. The same ANOVA followed by Tukey's approach would have been a robust one for gene expression/ qRT-PCR data.

Response: Thanks for your comments. The student's t test was used for gene expression analysis, since there are only two groups of treatments (SS and SulR). The ANOVA compares three or more groups. For RNAi experiments, we have at least three groups of treatment (DEPC control, dsGFP control, and the treatment of dsRNA), so ANOVA followed by Tukey's test was used.

7. Section 3.2: 3-4 fold is a modest increase in expression of UGT genes. Authors need to discuss the relevance of this modest increase in the discussion section. Specifically they need to discuss how much increase in UGT gene expression was reported in other studies/ other insect species.

Response: Thanks for your comments. The results of this modest increase expression of UGT genes were discussed in the discussion section, and similar results were compared with our results (Line 249-251).

8. Lines 244-246: What is the significance of CYP6AE gene cluster with the role of UGT's in resistance? This needs to be explained in detail in the discussion section.

Response: Thanks for your comments. We cited the reference of CYP6AE gene cluster was involved in detoxifying plant toxins and chemical insecticides just for indicating that a couple of UGT genes in a same subfamily (UGT344 subfamily) might participate in the detoxification of sulfoxaflor in A. gossypii. In the current reversion, this redundant sentence has been deleted.

Reviewer 2 Report

The manuscript titled “UDP-glycosyltransferases from the GUT344 family are involved in sulfoxaflor resistance in Aphis Gossypii Glover” investigated the mechanism of resistance, with a particular focus on UDP-glycosyltransferase (UGTs) genes in A. gossypii (the role of phase II metabolism) in the resistance of sulfoxaflor.  This includes the use of small molecules (sulfinpyrazone and 5-nitrouracil) to block UGT enzymes.  Blocking this metabolic pathway with these compounds resulted in synergism in the resistant strain of A. gossypii, but not in the susceptible strain. The investigators identified seven UGT genes, out of 15 studied, where over expressed. Suppression of three UGT genes restored toxicity of sulfoxaflor. 

Overall, the manuscript study is straight forward, the experiments appear to be well designed and performed and analyzed correctly; however, more statistical data should be added to Table 2. I have two major and several concerns:

MAJOR ISSUE/CONCERN:

  1. Plain and simple, the manuscript is not well written. The standard of grammar, punctuation (especially hyphens), the use of scientific and non-scientific terms, sentence structure, replacement of missing words, needs SIGNIFICANT attention. I have captured a lot of these changes that are needed below, but I am certain I have missed some.
  2. The authors have failed to convince me that UGTs are the MAIN metabolic resistance mechanism as outlined in the conclusions section. Could the two not be working together?

MINOR CONCERNS

Title: Aphis gossypii should be italicized.

Ln 14: rephrase start of sentence

Ln 15: Genus and species (even if first letter) needs to be italicized here, and throughout the rest of the manuscript.  These needs to be done for ALL species and genus’

Ln 20: general rule that a number under ten should be spelled out instead of using numeral

Ln 23: use of hyphen in this location is incorrect

Ln 32: insert “by” before virus

Ln 39-40: rephrase

Ln 43: Medicated? Incorrect spelling of mediated?

Ln 56: strike “the” and “time”

Ln 57: insert “the” before detoxification

Ln 57: the use of xenobiotics is incorrectly used here. The correct word is metabolites.  A xenobiotic is synthetically made substance/molecule. If the plant is producing the molecule it is really a metabolite, but in the context of a plosion it would be a toxin, which would still technically be incorrect within this context.

Ln 60: the use of a hyphen is not correct here

Ln 71: location of period is incorrect, and the sentence needs to be restructured.

Ln 73: the use of the hyphen is not correct

Ln 84: evaluate the use of your hyphens here AND THROUGHOUT THE MANUSCRIPT…THE ARE NOT BEING USED CORRECTLY.

LN 111: rephrase

Ln 143: the “-1” after the volume should be superscripted here and throughout the manuscript.

Ln 174: the “50” should be subscript here and throughout the manuscript

Ln 166-171 and Table 2: the authors state that some of the results are statistically significant; however, there is not statistical information provided on the significance level (only the method in a previous section). Also, the authors should consider the number of decimal points that are truly necessary for Table 2. More digits do not necessarily increase the accuracy of the results and the tables are very messy and difficult to read.

Ln 194:  GUT or UGT genes?

Author Response

The manuscript titled “UDP-glycosyltransferases from the GUT344 family are involved in sulfoxaflor resistance in Aphis Gossypii Glover” investigated the mechanism of resistance, with a particular focus on UDP-glycosyltransferase (UGTs) genes in A. gossypii (the role of phase II metabolism) in the resistance of sulfoxaflor.  This includes the use of small molecules (sulfinpyrazone and 5-nitrouracil) to block UGT enzymes.  Blocking this metabolic pathway with these compounds resulted in synergism in the resistant strain of A. gossypii, but not in the susceptible strain. The investigators identified seven UGT genes, out of 15 studied, where over expressed. Suppression of three UGT genes restored toxicity of sulfoxaflor. 

Overall, the manuscript study is straight forward, the experiments appear to be well designed and performed and analyzed correctly; however, more statistical data should be added to Table 2. I have two major and several concerns:

MAJOR ISSUE/CONCERN:

1. Plain and simple, the manuscript is not well written. The standard of grammar, punctuation (especially hyphens), the use of scientific and non-scientific terms, sentence structure, replacement of missing words, needs SIGNIFICANT attention. I have captured a lot of these changes that are needed below, but I am certain I have missed some.

Response: Thanks for your comments. All redundant hyphens, non-scientific terms, and similar mistakes were revised throughout the manuscript.

2. The authors have failed to convince me that UGTs are the MAIN metabolic resistance mechanism as outlined in the conclusions section. Could the two not be working together?

Response: Thanks for your comments. P450 is very important phase I metabolic enzyme and UGTs is phase Ⅱ metabolic enzyme, from our research (previous and this study) we could conclude that both P450 and UGTs are participated in the resistance process of A. gossypii against sulfoxaflor.

MINOR CONCERNS

Title: Aphis gossypii should be italicized.

Response: Thanks for your reminding. The scientific name of Aphis gossypii was written in italics throughout the manuscript.

Ln 14: rephrase start of sentence

Response: Thanks for your comments. The sentence was changed to “Our previous studies have demonstrated that UGTs participated in the detoxification of insecticides in Aphis gossypii” (Line 14).

Ln 15: Genus and species (even if first letter) needs to be italicized here, and throughout the rest of the manuscript.  These needs to be done for ALL species and genus’

Response: Thanks for your comments. The scientific names of insects were written in italics throughout the manuscript.

Ln 20: general rule that a number under ten should be spelled out instead of using numeral

Response: Thanks for your comments. The numeral number “7” was changed to “seven” (Line 20), and all numbers under ten were revised to correct format throughout the manuscript (Line 184, 188).

Ln 23: use of hyphen in this location is incorrect

Response: Thanks for your reminding, and the mistake was revised (Line 23).

Ln 32: insert “by” before virus

Response: Thanks for your comments. The word “by” was inserted between “indirectly” and “virus” (Line 32).

Ln 39-40: rephrase

Response: Thanks for your comments. The sentence was revised to “It is well known that insects have developed four types of resistance mechanisms to the chemical insecticides: metabolic resistance, target-site resistance, penetration resistance, and behavioral resistance” (Line 39-42).

Ln 43: Medicated? Incorrect spelling of mediated?

Response: Yes, mediated is correct, and the spelling mistake has been revised (Line 43).

Ln 56: strike “the” and “time”

Response: Thanks for reminding, the redundant words have been deleted (Line 57).

Ln 57: insert “the” before detoxification

Response: Thanks for your comments, and the word “the” has been insert before detoxification (Line 58).

Ln 57: the use of xenobiotics is incorrectly used here. The correct word is metabolites.  A xenobiotic is synthetically made substance/molecule. If the plant is producing the molecule it is really a metabolite, but in the context of a plosion it would be a toxin, which would still technically be incorrect within this context.

Response: Thanks for your comments. The incorrect use of “xenobiotics” has been revised to “metabolites” in the manuscript (Line 58).

Ln 60: the use of a hyphen is not correct here

Response: Thanks for your comments, and the hyphen was deleted (Line 62).

Ln 71: location of period is incorrect, and the sentence needs to be restructured.

Response: Thanks for your reminding. The redundant period after A. gossypii has been deleted (Line 72).

Ln 73: the use of the hyphen is not correct

Response: The hyphen has been deleted (Line 74).

Ln 84: evaluate the use of your hyphens here AND THROUGHOUT THE MANUSCRIPT…THE ARE NOT BEING USED CORRECTLY.

Response: Thanks for your comments. The hyphen has been deleted and all similar mistakes were revised throughout the manuscript (Line 86, 101, 261).

LN 111: rephrase

Response: Thanks for your comments. The sentence was revised to “To reveal the potential role of UGTs in metabolic resistance against sulfoxaflor, the synergistic effects of sulfinpyrazone (SUL) and 5-nitrouracil (5-NU) were investigated using the bioassay method described above” (Line 116-117).

Ln 143: the “-1” after the volume should be superscripted here and throughout the manuscript.

Response: Thanks for your comments. The formatting mistakes have been corrected throughout the manuscript (Line 124, 150-152, 159, 214-215).

Ln 174: the “50” should be subscript here and throughout the manuscript

Response: Thanks for your comments. “50” was wrote in subscript, and all similar mistakes have been revised throughout the manuscript (Line 113, 126-127, 159, 180, 220).

Ln 166-171 and Table 2: the authors state that some of the results are statistically significant; however, there is not statistical information provided on the significance level (only the method in a previous section). Also, the authors should consider the number of decimal points that are truly necessary for Table 2. More digits do not necessarily increase the accuracy of the results and the tables are very messy and difficult to read.

Response: Thanks for your comments. Table 2 has been revised and all number were kept two significant digits. In addition, the significance level was added in the text (Line 176).

Ln 194:  GUT or UGT genes?

Response: Thanks for reminding, and the spelling mistake of “UGT” was revised (Line 200).

Round 2

Reviewer 1 Report

Except for the english editing comment authors have addressed all the other suggestions. After thorough english and grammatical revision of this paper it should be ready for acceptance.

Author Response

Thanks for your comments. The entire manuscript has been professionally checked and edited for English usage, and some small mistakes were revised in this revised version.